# Store-Operated Calcium Entry Increases Nuclear Calcium in Adult Rat Atrial and Ventricular Cardiomyocytes

**DOI:** 10.3390/cells12232690

**Published:** 2023-11-23

**Authors:** Julia Hermes, Vesela Borisova, Jens Kockskämper

**Affiliations:** 1Institute for Pharmacology and Clinical Pharmacy, Biochemical and Pharmacological Centre (BPC) Marburg, University of Marburg, Karl-von-Frisch-Str. 2 K|03, 35043 Marburg, Germany; 2Department of Pharmacology and Clinical Pharmacology and Therapeutics, Medical University of Varna, Varna 9002, 55 Marin Drinov str., Bulgaria

**Keywords:** store-operated calcium entry, cardiomyocytes, atrial, ventricular, nuclear calcium, transient receptor potential canonical (TRPC), Orai

## Abstract

Store-operated calcium entry (SOCE) in cardiomyocytes may be involved in cardiac remodeling, but the underlying mechanisms remain elusive. We hypothesized that SOCE may increase nuclear calcium, which alters gene expression via calcium/calmodulin-dependent enzyme signaling, and elucidated the underlying cellular mechanisms. An experimental protocol was established in isolated adult rat cardiomyocytes to elicit SOCE by re-addition of calcium following complete depletion of sarcoplasmic reticulum (SR) calcium and to quantify SOCE in relation to the electrically stimulated calcium transient (CaT) measured in the same cell before SR depletion. Using confocal imaging, calcium changes were recorded simultaneously in the cytosol and in the nucleus of the cell. In ventricular myocytes, SOCE was observed in the cytosol and nucleus amounting to ≈15% and ≈25% of the respective CaT. There was a linear correlation between the SOCE-mediated calcium increase in the cytosol and nucleus. Inhibitors of TRPC or Orai channels reduced SOCE by ≈33–67%, whereas detubulation did not. In atrial myocytes, SOCE with similar characteristics was observed in the cytosol and nucleus. However, the SOCE amplitudes in atrial myocytes were ≈two-fold larger than in ventricular myocytes, and this was associated with ≈1.4- to 3.6-fold larger expression of putative SOCE proteins (TRPC1, 3, 6, and STIM1) in atrial tissue. The results indicated that SOCE in atrial and ventricular myocytes is able to cause robust calcium increases in the nucleus and that both TRPC and Orai channels may contribute to SOCE in adult cardiomyocytes.

## 1. Introduction

Cardiac excitation–contraction coupling is mediated by a transient rise in the cytoplasmic calcium (Ca) concentration and subsequent Ca decay, i.e., the Ca transient (CaT) [1]. During the depolarization of cardiomyocytes (ventricular and atrial), L-type Ca channels (LTCCs) open and Ca enters the cell with the subsequent release of Ca from the sarcoplasmic reticulum (SR) through ryanodine receptors (RyRs). This Ca increase causes contraction of the cell. Afterwards, cytosolic Ca is diminished by Ca reuptake into the SR via the sarco-/endoplasmic reticulum Ca-ATPase (SERCA) and by Ca extrusion from the cell via the sarcolemmal Na/Ca exchanger (NCX). Ca not only regulates cardiac contraction, it also plays a pivotal role in many other cellular processes, including metabolism and transcription [1]. The Ca-dependent regulation of transcription is mediated, in part, via alterations in nuclear Ca and has been termed excitation–transcription coupling [2].

Nuclear Ca in cardiomyocytes is regulated by both active and passive mechanisms. Ca may enter the nucleus passively by diffusion through nuclear pore complexes. Hence, each cytoplasmic CaT causes a delayed nuclear CaT [3,4]. In addition, IP_3_-dependent Ca release from the nuclear envelope may actively increase nuclear Ca [2,5]. Finally, SERCA activity limits the nuclear Ca increase in the systole [6]. Increases in nuclear Ca mediate altered transcription via Ca/calmodulin and CaMKII–HDAC–MEF2 and/or calcineurin–NFAT signaling, and this contributes to cardiac remodeling in hypertrophy and heart failure [2,7,8,9]. In the case of calcineurin–NFAT signaling, for example, it has been shown conclusively in adult cardiomyocytes that NFAT is localized and/or translocates to the nucleus in a Ca-, calcineurin-, and IP_3_-dependent manner—a mechanism that is facilitated in heart failure—and that increases in nuclear Ca as well as nuclear calcineurin acting as a Ca sensor are required in order to activate NFAT transcriptional activity [8,10,11].

In addition to the Ca-regulating proteins involved in excitation–contraction coupling, cardiomyocytes also express transient receptor potential canonical (TRPC) channels, small stromal interaction molecule1 (STIM1), and Orai. These proteins contribute to store-operated calcium entry (SOCE) in electrically non-excitable cells, a concept introduced by Putney and termed capacitive Ca entry at the time [12]. It holds that the depletion of intracellular Ca stores, i.e., the endoplasmic reticulum (ER), signals to the plasma membrane to open Ca-conducting channels for refilling of the stores. Current evidence suggests that STIM1 acts as Ca sensor in the ER membrane. Following Ca depletion of the ER, STIM1 can activate both Ca-selective Orai and non-selective TRPC cation channels in the plasma membrane mediating SOCE [13,14]. Orai and TRPC channels may even form heteromultimers in the plasma membrane activated by STIM and mediating SOCE [14,15,16,17]. In cardiomyocytes, there is a gap of knowledge regarding the existence, composition, and putative function of SOCE channels, and only a few studies have detected SOCE currents in cardiomyocytes. SOCE was first demonstrated in embryonic and neonatal cardiac cells [18] and, later, also in adult cardiomyocytes [19,20,21]. There is increasing evidence that the proteins involved in SOCE contribute to cardiac remodeling in hypertrophy and heart failure [22,23,24], rendering SOCE an attractive target for the treatment of cardiac disease. How SOCE might contribute to physiological or pathological Ca signaling in cardiomyocytes, however, remains elusive. We hypothesized that SOCE is able to increase nuclear Ca in adult cardiomyocytes, constituting a direct link to cardiac excitation–transcription coupling. Therefore, in the present study we established a protocol for eliciting and quantifying SOCE in adult cardiomyocytes and characterized the impact of SOCE on nuclear Ca signaling. SOCE pathways were dissected by pharmacological means. The results demonstrated that SOCE increases nuclear Ca with contribution from both TRPC and Orai channels. SOCE is considerably larger in atrial than in ventricular myocytes and associated with a greater expression of TRPC and STIM1 proteins in atrial myocardium.

## 2. Materials and Methods

### 2.1. Animals

Male Wistar-Kyoto rats (WKY, 250–350 g) at the age of 12–16 weeks were obtained from Janvier (Le Genest-Saint-Isle, France) and used for cell and tissue isolation. The animals were housed at a constant temperature of 22 °C and 55% humidity with ad libitum access to food and water. The experimental procedures used to isolate ventricular and atrial myocytes were approved by local animal welfare authorities at the University of Marburg and performed in accordance with European Union Council Directive 2010/63/EU and the German Animal Welfare Act (Tierschutzgesetz).

### 2.2. Isolation of Ventricular and Atrial Myocytes

Cardiomyocytes were isolated similar to protocols described previously [4,6]. All solutions were prepared from a Tyrode’s solution containing (mM) 130 NaCl, 5.4 KCl, 0.5 MgCl_2_, 0.33 NaH_2_PO_4_, 25 HEPES, and 22 glucose (pH = 7.4). Rats were anesthetized with isoflurane, weighed, and killed by decapitation. After opening of the chest, the heart was extracted rapidly and placed in oxygenated ice-cold cardioplegic solution. A cannula was inserted into the aorta and the vessels were flushed with a cannulation solution (Tyrode solution with 0.15 mM CaCl_2_ and 2 U/mL heparin). Subsequently, the heart was connected to the heated (37 °C) Langendorff perfusion system, where a Ca-free solution (0.4 mM EGTA, 10 mM 2,3-butanedione monoxime (BDM), and 2 U/mL heparin) was first applied followed by a modified Tyrode solution (solution with 0.2 mM CaCl_2_ and 10 mM BDM) with 0.6 mg/mL collagenase (CLS-2, Worthington, OH, USA) and 0.05 mg/mL type XIV protease to digest the tissue. After sufficient digestion time, the atrial tissue was separated from the ventricular tissue. Left and right atria were separated from each other and placed in an enzyme-free solution (containing 10 mM BDM, 2 mg/mL BSA, and 0.2 mM CaCl_2_) and cut gently to allow cell dissociation from the tissue. The cell suspension was placed on a rocking platform and Ca concentration was increased step by step using an adaptation protocol where solutions from 0.5 mM Ca up to 2 mM Ca were applied to reach a final concentration of 1.3 mM Ca.

The ventricular tissue was placed in an enzyme-free isolation solution (10 mM BDM, 2 mg/mL BSA and 0.5 mM CaCl_2_, room temperature (RT)), gently cut and filtered through sieve cloths with 300 µm pores (Kobe, Marburg, Germany). After sedimentation of the cells, the solution was changed to a 1 mM CaCl_2_ isolation solution (containing 2 mg/mL BSA), where they rested for another 15 min, before exchanging the solution for a 1.5 mM CaCl_2_ isolation solution. The cells were stored in this solution until use.

### 2.3. Confocal Linescan Imaging of Cytoplasmic and Nucleoplasmic CaTs

After isolation and Ca adaptation, the cells were plated on laminin (50 μg/mL)-coated glass-bottom dishes (WillCo-Wells, Amsterdam, The Netherlands) for 10 min. Cells were next loaded with the cell permeable fluorescent dye Fluo-4/AM (6.6 μM, ThermoFisher Scientific, Dreieich, Germany) for 20 min (ventricular) or for 25 min (atrial). Fluo-4/AM stock solutions (1 mM) were prepared in 20% (*w*/*v*) Pluronic^®^ F-127/DMSO (Invitrogen, Eugene, OR, USA). Cells were washed for another 20 (ventricular) or 25 (atrial) min and then bathed in a modified Tyrode solution containing (mM) 130 NaCl, 5 KCl, 1.5 CaCl_2_, 0.5 MgCl_2_, 10 HEPES, and 10 glucose; pH 7.4.

Subcellularly resolved CaTs were visualized using the linescan mode of a laser scanning confocal microscope (LSM 510, Carl Zeiss, Oberkochen, Germany) using an argon laser with an excitation wavelength of 488 nm and fluorescence emission at >505 nm and a 63× oil immersion objective lens (NA = 1.4). The scan line was set transversely across the cell, traversing the nucleus to visualize cytoplasmic and nucleoplasmic CaTs simultaneously [6,9]. Then, 1600 one-directional lines were recorded with a duration of 3.07 ms each at a depth of 12 bits. The pinhole was set to 140 μm resulting in a confocal plane of 1.0 μm thickness.

Ventricular myocytes were electrically stimulated at 1 Hz and atrial myocytes at 0.5 Hz at RT via two platinum electrodes (Myopacer, IonOptix, Amsterdam, The Netherlands). Representative linescan images (e.g., Figure 1) were edited with FIJI (ImageJ, Bethesda, MD, USA) by modifying the display range, which is provided in the respective figure legends. All linescan images were smoothed once. FIJI was also used for data analysis. Fluorescence traces were first background-subtracted. For data analysis, the following parameters were collected: diastolic and systolic fluorescence (F), resting fluorescence (F_rest_), amplitude (dF or ΔF), and SOCE fluorescence, i.e., the fluorescence value after the end of the SOCE protocol. The resting fluorescence F_rest_ was used for normalization.

### 2.4. Store-Operated Ca Entry: Experimental Protocol and Pharmacological Inhibitors Used

Fluo-4-loaded myocytes were electrically stimulated as described before. Stimulation was stopped for one minute and then switched on again to determine resting fluorescence, F_rest_, and the absence of spontaneous Ca waves during the stimulation pause and to confirm the presence of regular CaTs during electrical stimulation. Next, the solution was switched to a nominally Ca-free solution containing (mM) 130 NaCl, 5.4 KCl, 0.5 MgCl_2_, 10 HEPES, 10 glucose, 0.4 EGTA, adjusted to pH 7.4. Store depletion was achieved by adding thapsigargin (Biomol, Hamburg, Germany) (TG, 0.5 μM) to the solution. To exclude any contribution of L-type Ca channels or Ca influx via the NCX, verapamil (5 μM) and KB-R7943 (Cayman Chemical, Ann Arbor, MI, USA, 10 μM) were also added to this solution. The solution was administered continuously via a tube system for three minutes. Two bolus applications of caffeine (20 mM) were applied directly in the recording chamber to evoke acute store depletion. Each caffeine application was recorded by 8000 one-directional lines. Caffeine, at higher millimolar concentrations, leads to the opening of RyRs, which results in the release of stored Ca from the SR. Following store depletion and another minute of resting, the extracellular solution was switched to a 2 mM Ca solution containing (mM) 130 NaCl, 5.4 KCl, 2 CaCl_2_, 0.5 MgCl_2_, 10 HEPES, and 10 glucose, adjusted to pH 7.4, and cytosolic and nuclear fluorescence increases were recorded for 90 s.

The TRPC channel blocker SKF96365 (SKF, Cayman Chemical) and the Orai/CRAC channel inhibitors Synta 66 (S66, Sigma-Aldrich, Taufkirchen, Germany) and BTP-2 (Sigma-Aldrich) were used to study the impact of the respective channels on SOCE in cardiomyocytes. Cells were incubated during the de-esterification process with S66 or BTP-2, while SKF was applied acutely during the recording starting with the Ca-free solution (i.e., without any pre-incubation of the cells).

### 2.5. Detubulation of Ventricular Myocytes

Ventricular myocytes were isolated as described above. After loading the cells with Fluo-4 and subsequent washing, the cells were bathed in a solution containing (mM) 130 NaCl, 5.4 KCl, 1.5 CaCl_2_, 0.5 MgCl_2_, 10 HEPES, and 10 glucose, adjusted to pH 7.4. Detubulation of the ventricular myocytes occurred due to an osmotic shock, induced by treatment with 1.5 M formamide (Chemsolute, Renningen, Germany) for 15 min [25]. Afterwards, the cells were returned to the recording solution. Successful detubulation was confirmed by 2D confocal images with membrane staining using di-8-ANEPPS. After detubulation, the same SOCE protocol as described before was used to study SOCE in detubulated myocytes.

### 2.6. Confocal Imaging of T-Tubules

Untreated cells (control) and cells that had undergone the osmotic shock procedure were loaded with 10 μM di-8-ANEPPS (Cayman Chemical) for two minutes. After washing with recording solution for another two minutes, the cells were imaged using a confocal microscope in 2D mode with excitation and emission wavelengths of 488 nm and >505 nm, respectively.

### 2.7. Western Blots

Left ventricular (LV) and left atrial (LA) tissue of 12-week-old WKY were separated in ice-cold cardioplegic solution, rapidly frozen in liquid nitrogen, and stored at −80 °C. The tissues were homogenized manually using micro tissue grinders (Wheaton UK Limited, Rochdale, UK) and a mixture of protease and phosphatase inhibitors. Protein concentration of ventricular tissue lysates was measured using the Pierce BCA Protein assay kit (Thermo Fisher Scientific, Dreieich, Germany) in combination with a BSA standard curve. Protein expression of homogenates was determined using standard immunoblotting (Western blotting).

Twenty μg of total protein of four ventricular and four atrial samples were loaded on one 4–20% gradient gel (BioRad, Neuried, Germany) in an alternating pattern, every sample containing Laemmli buffer including 5% β-mercaptoethanol. A molecular weight marker was loaded on the first position of each gel. For each protein of interest, two gels with a total of eight ventricular and eight atrial samples were prepared. All samples were handled in an identical manner. Proteins were separated by polyacrylamide gel electrophoresis with voltage set to 90 V for 30 min, subsequently increasing to 120 V for 50 min. Proteins were transferred to nitrocellulose membranes (0.45 μm, BioRad, Germany) in a cooling chamber at 100 mA per gel for 1.5 h followed by 15 mA/gel overnight.

After protein transfer, membranes were cut between protein bands of interest to achieve simultaneous detection of differently sized proteins from one membrane. This step enables optimal use of the material and antibodies. For the analysis of TRPCs and STIM1, membranes were cut slightly below 55 kD. The upper parts of the membranes were used for TRPC or STIM1 detection and the lower parts for GAPDH detection. Membranes were then washed three times for 10 min with Tris-buffered saline Tween (TBST) and subsequently blocked for one hour at RT with 0.5% skimmed milk (skim milk powder) in TBST. After a second washing step, membranes were incubated with the primary antibody overnight at 4 °C. Primary antibodies were dissolved in 0.5% skimmed milk in TBST.

In this study, the following primary antibodies were used (company, catalogue number, dilution): rabbit anti-TRPC 1 (Alomone labs, Jerusalem, Israel, #ACC-010, 1:1000), rabbit anti-TRPC 3 (Alomone, #ACC-016, 1:1000), rabbit anti-TRPC 6 (Alomone, #ACC-017, 1:250), mouse anti-STIM1 (Thermo Scientific, #MA1-19451, 1:1000), and mouse anti-GAPDH (Calbiochem, Darmstadt, Germany, #CB1001, 1:50,000). GAPDH was used as a loading control.

Membranes were washed (TBST, 3× 10 min) and incubated with the corresponding secondary antibody (in 0.5% milk in TBST for one hour at RT) with subsequent washing with TBST. The following HRP-labeled secondary antibodies were used: goat anti-mouse IgG (Thermo Scientific, #31430, 1:5000) or goat anti-rabbit IgG (Thermo Scientific, #31460, 1:5000).

Chemiluminescence was detected using the Chemidoc-XRS system (BioRad, Germany). Membranes were incubated for one minute with HRP-Juice Plus (PJK GmbH, Kleinblittersdorf, Germany) or Super Signal West Femto (Thermo Scientific, Germany) reagents.

Blots were analyzed using FIJI (ImageJ, https://imagej.net/ij/) and the intensity of the protein bands was calculated and normalized to the respective loading control (GAPDH). Representative bands from ventricular and atrial tissue and their loading controls were cropped and are presented in the respective figure. The original immunoblots are shown in the Appendix A. Atrial bands were normalized to the averaged ventricular bands, set as 100%, and intensities from two membranes were analyzed together (eight samples for ventricles and atria).

### 2.8. Data and Statistical Analysis

Statistical analysis was performed using GraphPad Prism 9 (Version 9.0.2, GraphPad Software, Inc., San Diego, CA, USA). Cell data are presented as scatter dot plots and box plots indicating medians with Tukey’s whiskers. Immunoblot data are presented as scatter dot plots and bar graphs with means ± S.E.M. Individual *p*-values are shown in the figures. Differences are considered statistically significant when *p* < 0.05. Number of cells is provided as “*n*”, and number of animals as “*N*”.

Data sets were tested for normality with a D’Agostino and Pearson test, and when data were distributed normally, differences between two data sets were evaluated by paired or unpaired two-tailed Student’s *t*-test. Not normally distributed data were compared using Wilcoxon or Mann–Whitney tests. For multiple comparisons when multiple data sets were compared, variance analysis was conducted using the Kruskal–Wallis test with Dunn’s post-hoc. Groups were considered significantly different when *p* < 0.05. Correlation analysis was performed using Spearman’s test. In the case of significant correlation between two parameters, a linear regression analysis was conducted.

## 3. Results

### 3.1. Experimental Protocol for Eliciting SOCE in Cardiomyocytes

First, we established an experimental protocol for eliciting and quantifying SOCE in the cytosol and nucleus of adult cardiomyocytes. Figure 1A shows the protocol and Figure 1B the linescan images and corresponding normalized Fluo-4 (Ca) fluorescence traces of a ventricular myocyte at the times indicated (Figure 1B(a–j)). At the beginning (Figure 1B(a)), the myocyte was bathed in normal Tyrode’s (NT) solution and electrically stimulated (1 Hz). The resulting CaTs from cytosol (black) and nucleus (red) exhibited the characteristic pattern of a large and fast cytosolic CaT and a somewhat smaller, delayed nuclear CaT with a much slower decay [9]. These CaTs served for the later quantification of SOCE. Afterwards (Figure 1B(b)), stimulation was switched off and resting fluorescence (Frest) was recorded for the normalization of the fluorescence Ca signal. Stimulation was resumed (Figure 1B(c)) and CaTs reappeared. The solution was changed from NT to a zero-Ca solution containing thapsigargin, verapamil, and KB-R7943 to block SERCA, LTCC, and NCX, respectively. Under these conditions, CaTs were completely abolished (Figure 1B(d–f)). Two caffeine boli were applied (Figure 1B(g,h)) to probe for any remaining Ca in the SR. The first bolus (Figure 1B(g)) showed that the SR still contained Ca under these conditions. The second bolus (Figure 1B(h)) revealed that the SR was completely depleted of any Ca after the first bolus (see also Figure 2). Stimulation was switched off (Figure 1B(i)) and, finally, the cell was exposed to 2 mM Ca NT solution (Figure 1B(j)). This caused a slow but steady increase in cytosolic and nuclear Ca, which was taken as an indication of SOCE. The amplitude of this slow Ca increase at the end of the 90 s recording (Figure 1B(j)) served as a measure of SOCE. The exact fluorescence values used for the quantification of SOCE were taken from an additional linescan recording conducted a few seconds after the 90 s recording shown here (as detailed in Appendix A). The protocol for eliciting SOCE as shown in this figure and Appendix A was used in the entire study.

In order to elicit SOCE reliably, the SR should be depleted of any Ca. We evaluated this by comparing the amplitudes of the cytosolic and nuclear Ca increases induced by the two caffeine applications. Figure 2A shows linescan images from a ventricular myocyte during the two caffeine applications (same cell and images as in Figure 1B). Summarized data from 39 ventricular myocytes are presented in Figure 2B. They demonstrate that the first caffeine bolus (#1, left) elicited large Ca increases in the majority of cells. Only four cells were entirely Ca-depleted under these conditions. The second caffeine bolus (#2, right), however, never elicited any detectable Ca increase, indicating that, following the first bolus, the SR was Ca-depleted and not capable of re-gaining any Ca.

### 3.2. SOCE Characteristics in the Cytosol and Nucleus of Ventricular Cardiomyocytes

Figure 3 displays the characteristics of SOCE in the cytosol and nucleus of ventricular myocytes. CaT traces at the beginning and SOCE traces at the end of the protocol from an example cell are illustrated in Figure 3A. Clearly, SOCE is smaller than the electrically stimulated CaTs, both in the cytosol and nucleus. The CaT amplitudes were larger in the cytosol (Figure 3B), whereas the SOCE amplitudes were larger in the nucleus (Figure 3C). When the SOCE amplitudes were normalized to the CaT amplitudes in the same cell, they amounted to ≈15% in the cytosol and ≈25% in the nucleus, with a few cells displaying much higher values (Figure 3D). There was a direct linear correlation between the SOCE amplitude in the cytosol and the nucleus (Figure 3E), suggesting that the nuclear Ca increase during SOCE depends on the cytosolic Ca increase. When the nuclear Ca increases were normalized to the cytosolic Ca increases, these ratios amounted to ≈0.6–0.7 for CaTs and ≥1 for SOCE (Figure 3F). Thus, for electrically stimulated CaTs, the nuclear Ca increases are smaller than the cytosolic increases, whereas during SOCE, the nuclear Ca increase is equally large as that in the cytosol. The data show that ventricular myocytes exhibit SOCE with a robust Ca increase in the nucleus.

### 3.3. TRPC and Orai Channels Contribute to SOCE in Ventricular Myocytes

In order to test which mechanisms might contribute to SOCE in ventricular myocytes, inhibitors of TRPC and Orai channels were used. A shown in Figure 4, S66 (1 μM) and BTP-2 (3 μM), two different inhibitors of Orai [26,27], reduced SOCE (normalized to the CaT) by ≈33% and ≈33% in the cytosol (left) and by ≈44% and ≈38% in the nucleus, respectively (right, Figure 4B). SKF96365 (5 μM), a non-selective and widely used inhibitor of TRPC channels [28], also reduced SOCE by a similar degree, i.e., by ≈36% in the cytosol and by ≈45% in the nucleus (Figure 4B). The combination of S66 and SKF96365 had an even greater effect, reducing SOCE by ≈67% in the cytosol and by ≈64% in the nucleus (Figure 4B). The results suggested that both TRPC and Orai channels contribute to the cytosolic and nuclear Ca increase during SOCE in ventricular myocytes.

### 3.4. SOCE Persists in Ventricular Myocytes after Detubulation

A regular T-tubular system in ventricular myocytes serves to conduct the action potential deep into the cell. The T-tubules are enriched with LTCC to trigger SR Ca release via RyR in a synchronized manner throughout the myocyte [29]. Previous studies have shown that T-tubules may also contain TRPC proteins [30]. Therefore, we tested whether the detubulation of ventricular myocytes might reduce SOCE either in the cytosol and/or the nucleus. The latter scenario seems plausible, because Ca entering the cell through TRPC (and Orai) channels in the peripheral sarcolemma must travel larger distances to reach the nucleus than Ca entering from T-tubules, which form a cage around the nucleus [31]. For detubulation, ventricular myocytes were exposed to formamide-induced osmotic shock [25]. The results are presented in Figure 5. Figure 5A shows results from untreated control ventricular myocytes. An example cell stained with di-8-ANEPPS contained a regular T-tubular system (left). A different control cell loaded with Fluo-4 (same cell as in Figure 3A) displayed regular electrically stimulated cytosolic and nuclear CaTs (middle) and robust SOCE both in the cytosol and nucleus (right). Formamide treatment resulted in the removal of T-tubules (Figure 5B, left). A ventricular myocyte detubulated in this way exhibited large reduction in both cytosolic and nuclear CaTs (middle), as described previously for cytosolic CaTs [32], because LTCCs are concentrated in the T-tubules. By contrast, the detubulated cell still exhibited robust SOCE both in the cytosol and nucleus (right). Summarized data are shown in Figure 5C–E. The CaT amplitude was reduced in detubulated ventricular myocytes from ≈2.9 ΔF/Frest to ≈1.9 ΔF/Frest (or by 66%) in the cytosol and from ≈1.8 ΔF/Frest to ≈1.1 ΔF/Frest (or by 61%) in the nucleus (Figure 5C). In sharp contrast, the SOCE amplitudes did not differ between ventricular myocytes with or without T-tubules, neither in the cytosol nor in the nucleus (Figure 5D). Thus, when the SOCE amplitudes were normalized to the CaT amplitudes, detubulated cells exhibited a much greater relative SOCE both in the cytosol and in the nucleus (Figure 5E).

### 3.5. SOCE in the Cytosol and Nucleus of Atrial Cardiomyocytes

Atrial myocytes exhibit some distinct differences compared with ventricular myocytes with regard to morphology and function, including excitation–contraction coupling [33]. Hence, we also characterized SOCE in the cytosol and nucleus of atrial myocytes. To this end, we used the same experimental protocol as for ventricular myocytes (compare Figure 1). As shown in Figure 6A, atrial myocytes did not display a regular T-tubular system (left), but rather large electrically stimulated cytosolic and nuclear CaTs (middle) as well as readily detectable SOCE in the cytosol and nucleus (right). The CaT amplitudes amounted to ≈3.2 ΔF/Frest in the cytosol and to ≈1.4 ΔF/Frest in the nucleus (Figure 6B). The SOCE amplitudes, on the other hand, did not differ between cytosol and nucleus amounting to ≈1.0–1.1 ΔF/Frest in both compartments (Figure 6C). The relative SOCE amplitudes (normalized to CaT amplitudes) were larger in the nucleus (Figure 6D). There was a direct linear correlation between the SOCE amplitudes in the cytosol and the nucleus (Figure 6E). Finally, the ratio of the nuclear to cytosolic amplitude amounted to ≈0.5 for the electrically stimulated CaT and to ≈1 for SOCE (Figure 6F). These properties of cytosolic and nuclear SOCE in atrial myocytes were very similar to the ones in ventricular myocytes (compare Figure 3). A notable difference was that SOCE (in the cytosol and nucleus) was greater in atrial than in ventricular myocytes, both in absolute (ΔF/Frest) as well as relative (SOCE/CaT) terms (compare Figure 3C,D).

### 3.6. TRPC and Orai Channels Contribute to SOCE in Atrial Myocytes

In order to estimate the contribution of TRPC and Orai channels to SOCE in atrial myocytes, we used the same pharmacological approach as for ventricular myocytes, i.e., S66 or BTP-2 (Orai inhibitors), SKF96365 (TRPC inhibitor) and a combination of S66 and SKF. Figure 7 shows the results for individual cells (A) and summarized data (B). All four treatments caused a large reduction in SOCE both in the cytosol and nucleus: S66 reduced cytosolic and nuclear SOCE by ≈70% and ≈79%, BTP-2 by ≈79% and ≈79%, and SKF96365 by ≈79% and ≈75%, respectively. Interestingly, combining S66 and SKF did not increase the inhibitory effect on SOCE, neither in the cytosol (−≈67%) nor in the nucleus (−≈69%). The results suggested that TRPC as well as Orai channels contribute to SOCE in atrial myocytes.

### 3.7. Expression of TRPC Proteins and STIM1 Is Higher in Atrial Myocardium

Because both atrial and ventricular myocytes exhibited robust SOCE, and SOCE was greater in atrial than in ventricular myocytes, we compared the expression of the proteins presumably involved in SOCE between atrial and ventricular myocardia. The results are presented in Figure 8. Original immunoblots are shown in A and summarized data in B. Clearly, the expression of major TRPC isoforms as well as STIM1 was higher in atrial myocardium by a factor of ≈1.4 (STIM1), ≈2.0 (TRPC6), ≈2.5 (TRPC1), and ≈3.6 (TRPC3). The higher expression of these proteins in the atrial myocardium matches the greater SOCE observed in the atrial myocytes.

## 4. Discussion

The role of SOCE in electrically non-excitable cells has been studied widely, and various physiological and pathological functions of SOCE are known [34,35,36]. In cardiomyocytes, however, the existence and potential relevance of SOCE remains elusive. Here we show that both atrial and ventricular cardiomyocytes from adult rat hearts exhibit SOCE following the depletion of SR Ca stores. A regular T-tubular system is not required for this SOCE and both TRPC and Orai channels may contribute to sarcolemmal Ca entry. Importantly, SOCE into the cytosol is transmitted readily into the nucleus in both atrial and ventricular myocytes, suggesting a link between SOCE and Ca-dependent regulation of transcription and, hence, cardiac remodeling.

In this study, we established a unique experimental protocol to elicit and quantify SOCE in the cytosol and nucleus of single cardiomyocytes. The key features of this protocol include (1) the determination of electrically stimulated cytosolic and nuclear CaTs followed by (2) the suppression of electrically stimulated CaTs in zero-Ca solution (plus inhibitors of LTCC, NCX and SERCA) and the complete depletion of SR Ca (as mediated and verified by two caffeine applications); finally, (3) the re-addition of 2 mM Ca elicited a slowly developing SOCE both in the cytosol and nucleus. Because CaTs and SOCE were determined in the same cell, the SOCE amplitudes could be quantified with respect to the CaT amplitudes. Previously, the in situ calibration of cytosolic and nuclear CaTs in rat ventricular myocytes yielded CaT amplitudes in the range of ≈700–1200 nM for the cytosol and ≈400–900 nM for the nucleus, respectively [4,9]. Based on these numbers, we estimate that the SOCE amplitudes observed here translate into Ca increases of ≈100–180 nM in the cytosol and of ≈100–220 nM in the nucleus. These SOCE amplitudes are small compared to the respective CaT amplitudes in the same cell, but still robust and likely capable of increasing the activity of Ca-dependent enzymes involved in excitation–transcription coupling, i.e., calcineurin and CaMKII. The direct linear correlation between cytosolic and nuclear SOCE suggests that, during SOCE, the nuclear Ca increase is caused by the cytosolic Ca increase, presumably by Ca diffusion through nuclear pore complexes, as shown previously for electrically stimulated CaTs [4,6].

Reports on SOCE or I_CRAC_ in adult ventricular myocytes are scarce; however, previous studies have demonstrated the existence of SOCE/I_CRAC_ in adult rat and mouse ventricular myocytes [19,20,21,22,24,37,38,39,40]. Some studies, however, have failed to detect a measurable SOCE or found SOCE only in a small subset of ventricular myocytes in the absence of hypertrophy [24,41,42]. The reason for this discrepancy remains unclear. It may be related, at least in part, to the experimental protocol employed to elicit SOCE. Furthermore, the inhibition of SERCA alone (by thapsigargin or CPA) may not be sufficient to completely deplete the SR. Wen et al. [21] showed that maximal SOCE activation in cardiomyocytes requires the complete depletion of SR Ca by both the inhibition of SERCA (by thapsigargin or CPA) and caffeine-induced release of SR Ca. Consistent with this notion, we found that in the presence of thapsigargin and in the absence of Ca influx and electrically stimulated CaTs, the majority of ventricular myocytes still contained large amounts of caffeine-releasable Ca, and that one caffeine bolus was required to completely deplete the SR of Ca (Figure 2). Here, we were able to reliably detect a slowly developing SOCE in adult rat ventricular myocytes, both in the cytosol and nucleus of the cells. The magnitude of this SOCE was quite variable ranging from less than 10% up to ≈75% of the cytosolic CaT amplitude (Figure 3). Based on pharmacological data, we suggest that this SOCE was mediated by the activation of both TRPC (SKF-sensitive) and Orai (BTP-2- and S66-sensitive) channels (Figure 4). Plasma membrane channels mediating SOCE may be TRPC hetero- or homomers, Orai homomers, or even TRPC:Orai heteromers [14,15,16,17]. The additive effect of SKF and S66 might suggest that both TRPC multimers as well as Orai multimers were involved in the SOCE in ventricular myocytes. The fact that nuclear and cytosolic SOCE were inhibited to the same degree by the various blockers again argues for the notion that the nuclear Ca increase is mediated by the cytosolic Ca increase.

The kinetics of cytosolic and nuclear Ca increases mediated by SOCE are very slow, in particular when compared to the kinetics of CaTs during normal excitation–contraction coupling. Furthermore, it is still unclear whether SOCE can be activated in cardiomyocytes during normal excitation–contraction coupling. Direct measurements of SR Ca in adult cardiomyocytes indicate that intraluminal free SR Ca in the diastole amounts to ≈1–1.5 mM and that SR Ca declines by 24% to 63% in the systole, leaving residual SR Ca in the range of at least a few hundred micromolar [43]. Thus, it appears unlikely that complete SR Ca depletion occurs under physiological conditions and that high SOCE may be activated in cardiomyocytes. Nevertheless, some SOCE channels, in particular TRPC channels, can be activated by other means, e.g., stretching. Moreover, a recent study demonstrated that elevated extracellular Ca could induce a background Ca influx carried by TRPC6 channels in ventricular myocytes [44]. The Ca entering the cell via TRPC6 channels led to elevations of diastolic Ca and could be taken up into the SR by SERCA to increase SR Ca load and electrically stimulated CaTs and, eventually, to cause Ca waves [44]. Based on these findings, we suggest that, if SOCE, i.e., a slow but persistent background Ca influx via Orai and/or TRPC channels, was activated in cardiomyocytes, then this would lead to a similar sequence of events. Since the SR and the nuclear envelope are one highly interconnected Ca-storing organelle [45], increases in SR Ca load would be readily transmitted to the nuclear envelope. Under these conditions, increased diastolic Ca and CaTs in the cytosol are also likely to be transferred into the nucleus (via diffusion through nuclear pores and elevated nuclear envelope Ca stores), resulting in elevated nuclear diastolic Ca and CaTs.

T-tubules are indispensable for normal excitation–contraction coupling in adult ventricular myocytes as they are highly enriched with LTCC, which functionally couple to adjacent RyR2 from the junctional SR, thereby enabling synchronous SR Ca release throughout the myocyte [29]. Moreover, T-tubules form a cage around the nucleus with potential ramifications for nuclear Ca signaling [31]. It is not known, however, whether T-tubules are also involved in the regulation of SOCE. If the ion channels mediating SOCE were enriched in T-tubules, then Ca entering the cell through these channels would have preferential access to the cell center and the nucleus. Hence, we addressed this issue by comparing SOCE in normal versus detubulated ventricular myocytes (Figure 5). Successful detubulation was confirmed by the visualization of the (lack of) T-tubules and by greatly diminished electrically stimulated CaTs. The SOCE amplitudes both in the cytosol and nucleus, however, were essentially unchanged in detubulated cells. Thus, SOCE and the SOCE-induced nuclear Ca increase do not require T-tubules, suggesting that the ion channels mediating SOCE, presumably both TRPC and Orai channels, are predominantly distributed in the non-T-tubular sarcolemma. How does this compare to the subcellular localization of the proteins involved in SOCE? In adult ventricular myocytes, STIM1 is found in a regular striated pattern at the Z disks, in a perinuclear compartment, close to the peripheral sarcolemma and at the intercalated disks [27,37,40,42]. By contrast, Orai1 is found predominantly at the peripheral sarcolemma and in a weaker, non-striated, punctate staining throughout the myocyte [22,37]. Finally, TRPC channels may occur in a striated pattern, in particular TRPC1, 3, and 6, but mostly more prominently at the peripheral sarcolemma and/or the intercalated disks [30,37,40,46]. Thus, both Orai1 and (some) TRPC channels exhibit prominent expression at the peripheral sarcolemma, which might explain why we did not observe reduced SOCE after T-tubule disruption, neither in the cytosol nor in the nucleus. The latter finding also argues against the preferential access of SOCE to the nucleus, e.g., via the localization of SOCE proteins in the T-tubules forming a cage around the nucleus.

Ventricular and atrial myocytes represent distinct types of cardiomyocytes exhibiting various differences with regard to morphology, ultrastructure, and expression of ion channels (to name but a few), which result in differences in action potential, excitation–contraction coupling, subcellular Ca handling, and contractile activity [33,47,48,49]. Here we extend these differences between atrial and ventricular cardiomyocytes to SOCE and the expression of the underlying proteins. Similar to ventricular myocytes, atrial myocytes exhibited robust SOCE both in the cytosol and nucleus, and sarcolemmal Ca entry was inhibited by SKF96365, BTP-2, and S66, suggesting that both TRPC and Orai channels contribute to SOCE. The SOCE-mediated Ca increases in the cytosol and nucleus, however, were ≈two-fold larger in atrial myocytes. Based on previous calibrations and considering that atrial myocyte CaTs are as large as ventricular myocyte CaTs [4,9], we estimate that SOCE increased cytosolic and nuclear Ca by ≈210–360 nM and ≈250–580 nM, respectively, in atrial myocytes. Furthermore, direct comparison between atrial and ventricular myocardium revealed a much higher expression of TRPC and STIM1 proteins in the atrial myocardium (by a factor of ≈1.4 to 3.6). This suggests a more prominent role for SOCE (mediated by the STIM–Orai axis and TRPC channels) in atrial myocytes as compared with ventricular myocytes.

## 5. Conclusions

In conclusion, this study has demonstrated that SOCE can increase cytosolic and nuclear Ca concentration considerably in both atrial and ventricular cardiomyocytes. SOCE is mediated by TRPC and Orai channels located predominantly in the non-T-tubular surface sarcolemma. The SOCE-mediated increase in nuclear Ca provides a potential mechanistic link between SOCE proteins and cardiac remodeling.

## Figures and Tables

**Figure 1 cells-12-02690-f001:**
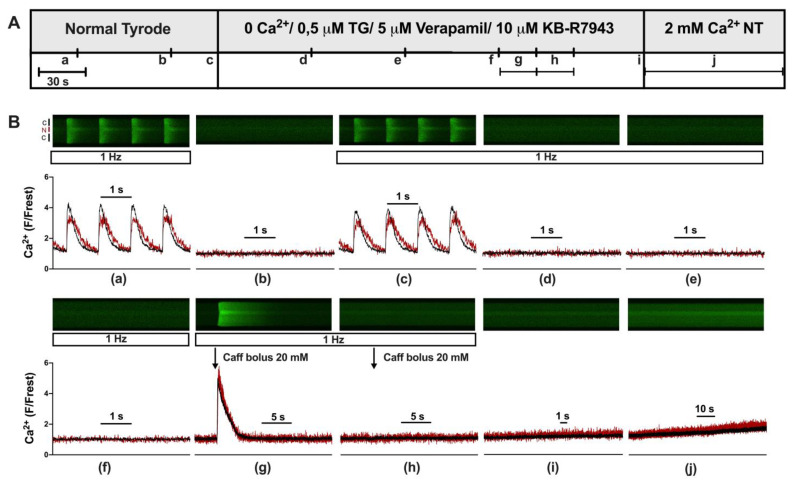
Experimental protocol used to elicit store-operated calcium entry in ventricular myocytes. (**A**) Experimental protocol showing the various solutions used to elicit store-operated calcium entry (SOCE). (**B**) Representative linescan images of cytoplasmic (black) and nuclear (red) Ca at 1 Hz stimulation (**a**,**c**–**h**) or without stimulation (**b**,**i**,**j**) from a WKY ventricular myocyte with corresponding normalized averaged fluorescence traces. Scan line was positioned perpendicular to the longitudinal axis of the cell traversing one nucleus, allowing for simultaneous recordings of cytosolic (black) and nuclear (red) Ca. Cells were first electrically stimulated at 1 Hz resulting in cytosolic and nuclear CaTs (**a**). After a resting period (**b**), the stimulation was turned on again (**c**) and NT was switched to a 0 Ca solution with 0.5 μM thapsigargin (TG), 5 μM verapamil, and 10 μM KB-R7943 resulting in elimination of CaTs (**d**–**f**). Twenty mM caffeine was applied twice (**g**,**h**) to empty intracellular Ca stores. After another short resting period (**i**), a 2 mM Ca (NT) solution was applied for 90 s to evoke store-operated calcium entry (**j**). The display range (after background correction) was 0–1200 for linescan images (**a**,**c**,**g**) and 0–800 for all other linescan images.

**Figure 2 cells-12-02690-f002:**
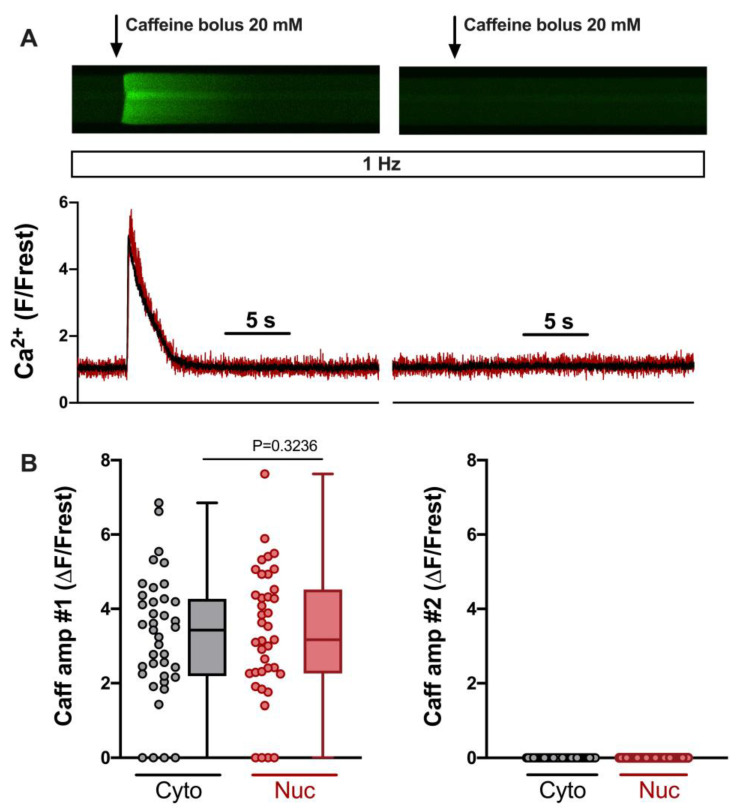
SR Ca load in 0 Ca/TG/verapamil/KB-R7943 solution determined by two caffeine bolus applications. (**A**) Representative linescan images of Fluo-4 fluorescence caffeine transients in a field-stimulated (1 Hz) ventricular myocyte from WKY during the first (**left**) and second (**right**) caffeine bolus application and corresponding normalized fluorescence traces (**below**), cytosolic (black) and nuclear (red) traces are shown. Same cell and images as in Figure 1. (**B**) Averaged values of cytosolic (grey) and nuclear (red) first caffeine amplitude (**left**) and second caffeine amplitude (**right**). Wilcoxon test; *p*-value as indicated; *n* = 39 cells from *N* = 16 rats. The display range (after background correction) was 0–1200 for both linescan images.

**Figure 3 cells-12-02690-f003:**
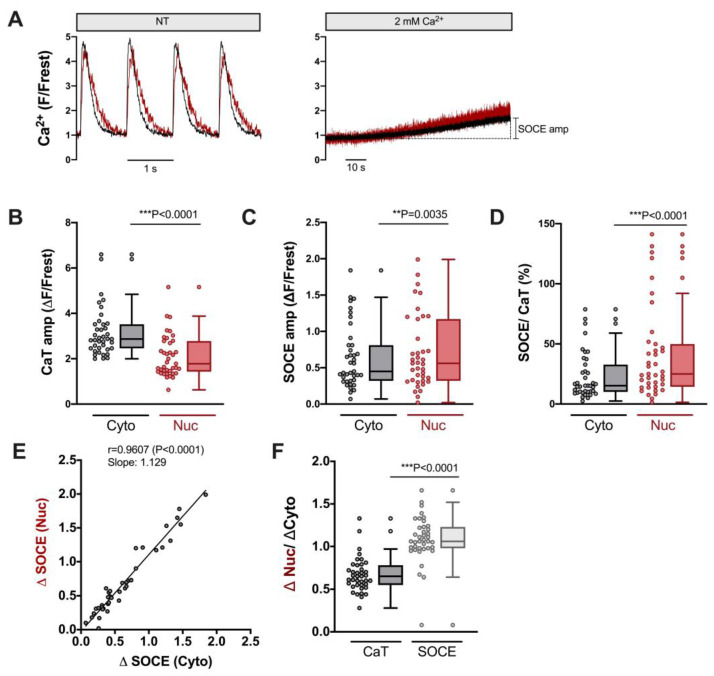
Robust SOCE in cytoplasm and nucleus of ventricular cardiomyocytes. (**A**) Representative normalized fluorescence traces of CaTs at 1 Hz stimulation (**left**) and SOCE evoked by re-addition of 2 mM Ca (**right**) of a ventricular myocyte. Cytosolic (black) and nuclear (red) traces are shown. (**B**) Averaged values of CaT amplitudes (ΔF/Frest). (**C**) Averaged values of SOCE amplitudes (ΔF/Frest). (**D**) SOCE amplitudes in the cytoplasm and nucleus normalized to the respective CaT in the very same cell. Cytosolic (Cyto, grey) and nuclear (Nuc, red) data is shown. (**E**) Linear correlation of nuclear and cytosolic SOCE amplitudes (Spearman correlation). (**F**) Average values of nuclear/cytosolic ratios of CaT amplitudes (black) or SOCE amplitudes (grey). Wilcoxon/Mann–Whitney test; *p*-values as indicated; **, *p* < 0.01; ***, *p* < 0.001; *n* = 39 cells from *N* = 16 rats.

**Figure 4 cells-12-02690-f004:**
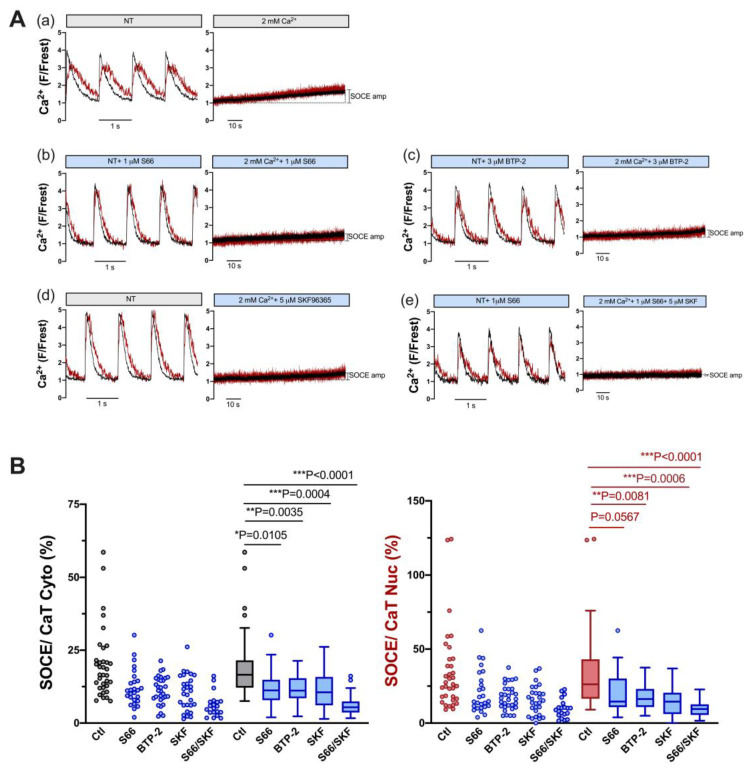
SOCE in ventricular myocytes is reduced by pharmacological inhibitors of TRPC and Orai channels. (**A**) Representative normalized fluorescence traces of CaTs at 1 Hz stimulation (**left**) and normalized fluorescence traces of SOCE (**right**) of an untreated control myocyte in the presence of DMSO (**a**) and of ventricular myocytes measured in the presence of 1 μM S66 (**b**), 3 μM BTP-2 (**c**), 5 μM SKF96365 (**d**), or 1 μM S66 plus 5 μM SKF96365 (**e**). Cytosolic (black) and nuclear (red) traces are shown. (**B**) Averaged values of SOCE/CaT (%) for cytoplasm (**left**) and nucleus (**right**) of untreated control cells (ctl) and of cells treated with the various inhibitors, as indicated, depicted as scatter dot plots and box plots with medians. DMSO ctl: *n* = 35, *N* = 17; S66: *n* = 26, *N* = 4; BTP-2: *n* = 31, *N* = 5; SKF: *n* = 28, *N* = 5; S66/SKF: *n* = 20, *N* = 3; Kruskal–Wallis test with Dunn’s multiple comparisons test; *p*-values as indicated; *, *p* < 0.05; **, *p* < 0.01; ***, *p* < 0.001.

**Figure 5 cells-12-02690-f005:**
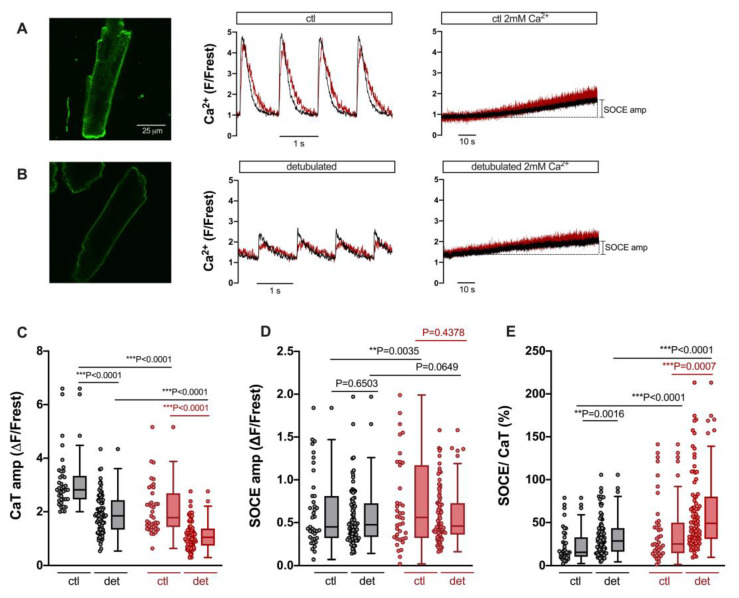
SOCE persists after detubulation of ventricular myocytes, but CaT is decreased. (**A**) Representative 2D confocal image (**left**) of a ventricular myocyte stained with the cell membrane dye di-8-ANEPPS (10 μM). Normalized fluorescence traces of CaTs at 1 Hz stimulation (**middle**) and SOCE (**right**) of a different cell (same cell as shown in Figure 3A). Cytosolic (black) and nuclear (red) traces are shown. (**B**) Representative 2D confocal image (**left**) of a detubulated ventricular myocyte stained with di-8-ANEPPS. Normalized fluorescence traces of CaTs at 1 Hz stimulation (**middle**) and SOCE (**right**) of a different detubulated cell. The display range for the images in (**A**) and (**B**) was 250–3000. (**C**) Averaged values of CaT amplitudes in cytoplasm (grey) and nucleus (red), (**D**) SOCE amplitudes, and (**E**) SOCE amplitudes normalized to the respective CaT in the very same cell in control (ctl) or detubulated (det) ventricular myocytes. Wilcoxon/Mann–Whitney test; ctl: *n* = 39, *N* = 16 (same cells and data as in Figure 3); detubulated: *n* = 84, *N* = 12; *p*-values as indicated; **, *p* < 0.01; ***, *p* < 0.001.

**Figure 6 cells-12-02690-f006:**
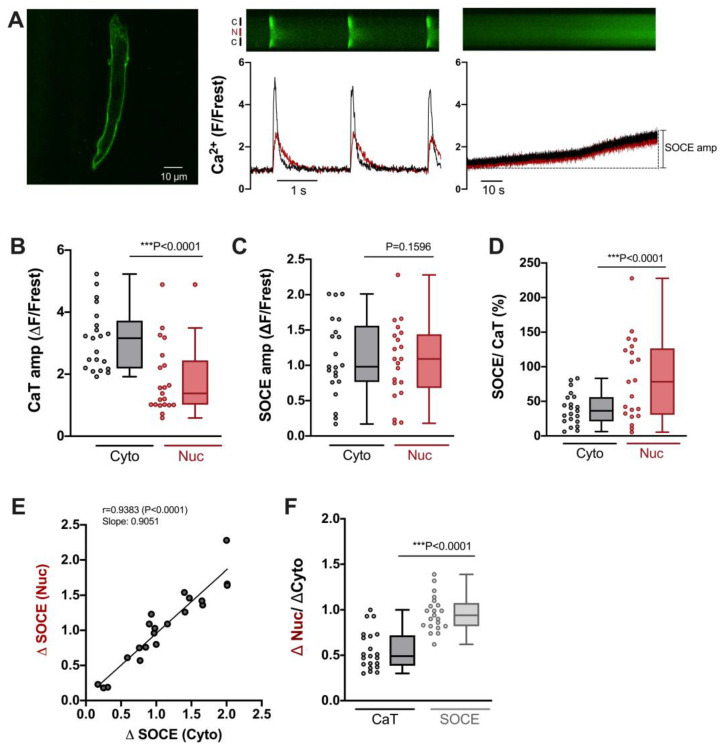
Characteristics of SOCE in atrial cardiomyocytes. (**A**) Representative 2D confocal image (**left**) of an atrial myocyte; cell membrane was stained with di-8-ANEPPS (10 μM). Display range was 280–2300. Normalized fluorescence traces (of a different cell) of CaTs at 0.5 Hz stimulation (**middle**) and SOCE (**right**) with corresponding original linescan images shown above. Display range (after background correction) for both linescan images was 0–800. Cytosolic (black) and nuclear (red) traces are shown. (**B**) Averaged values of CaT amplitudes (ΔF/F_rest_), (**C**) SOCE amplitudes (ΔF/F_rest_), and (**D**) SOCE amplitudes normalized to the respective CaT amplitude in the very same cell. Data for cytosol (Cyto, grey) and nucleus (Nuc, red) is shown. (**E**) Linear correlation of nuclear and cytosolic SOCE amplitudes (Spearman correlation). (**F**) Average values of nuclear/cytosolic ratios of CaT amplitudes (black) or SOCE amplitudes (grey). Wilcoxon test; *n* = 21 cells from *N* = 5 rats; *p*-values as indicated; ***, *p* < 0.001.

**Figure 7 cells-12-02690-f007:**
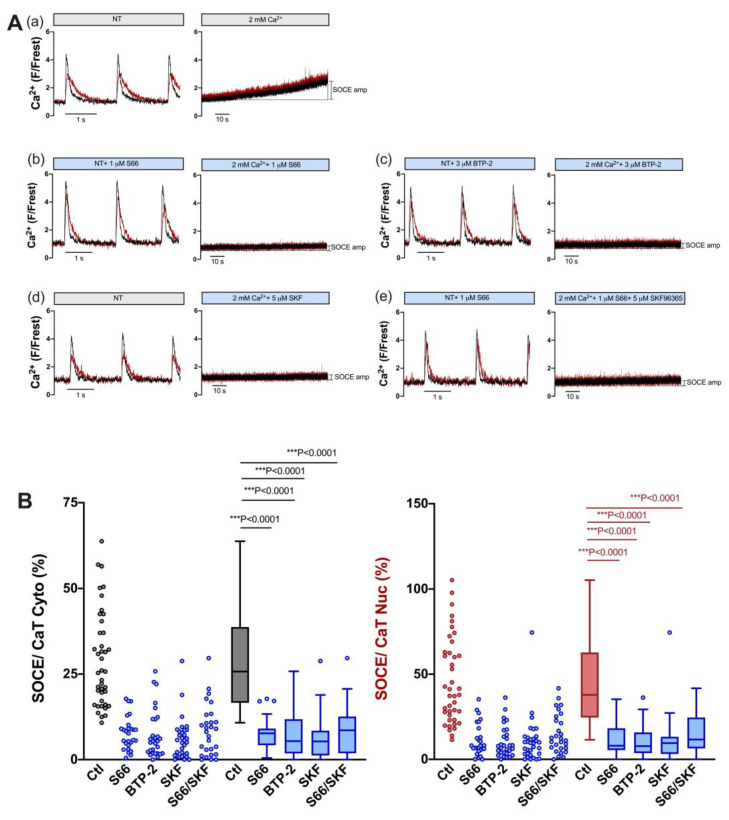
SOCE in atrial myocytes is reduced by pharmacological inhibitors of TRPC and Orai channels. (**A**) Representative normalized fluorescence traces of CaTs at 0.5 Hz stimulation (**left**) and normalized fluorescence traces of SOCE (**right**) of an untreated control atrial myocyte in the presence of DMSO (**a**) and of atrial myocytes measured in the presence of 1 μM S66 (**b**), 3 μM BTP-2 (**c**), 5 μM SKF96365 (**d**), or 1 μM S66 plus 5 μM SKF96365 (**e**). Cytosolic (black) and nuclear (red) traces are shown. (**B**) Averaged values of SOCE/CaT (%) for cytoplasm (**left**) and nucleus (**right**) of untreated control cells (ctl) and of cells treated with the various inhibitors, as indicated, depicted as scatter dot plots and box plots with medians. Kruskal–Wallis test with Dunn’s multiple comparisons test; DMSO ctl: *n* = 41, *N* = 17; S66: *n* = 27, *N* = 5; BTP-2: *n* = 29, *N* = 4; SKF: *n* = 32, *N* = 4; S66/SKF: *n* = 28, *N* = 7; *p*-values as indicated; ***, *p* < 0.001.

**Figure 8 cells-12-02690-f008:**
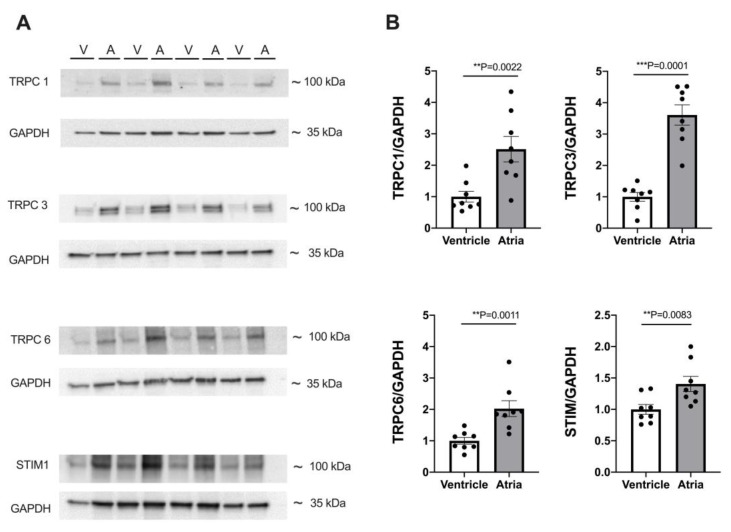
Expression of TRPC and STIM proteins differs between atrial and ventricular tissue. (**A**) Representative immunoblots of TRPC1, 3, 6, and STIM1 with housekeeping protein GAPDH in ventricular (V) and atrial (A) tissue. (**B**) Averaged data of expression of TRPC1, 3, 6, and STIM1 in atrial and ventricular tissue from 8 WKY hearts. Protein expression was normalized to GAPDH expression and ventricular data served as reference (=100%). Student’s *t*-test; *p*-values as indicated; **, *p* < 0.01; ***, *p* < 0.001.

## Data Availability

Data are contained within the article or Appendix A. Original data sets not shown here are available from the corresponding author upon reasonable request.

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
