# Peer review of "Store-Operated Calcium Entry Increases Nuclear Calcium in Adult Rat Atrial and Ventricular Cardiomyocytes"

_cells, 2023, doi:10.3390/cells12232690_

Round 1
Reviewer 1 Report
Comments and Suggestions for Authors
Measuring calcium currents in cardiomyocytes remains challenging. In this paper, the authors proposed an elegant way to measure SOCE currents. The advantage of the method is that SOCE currents and Ca transients are measured in a same cell, which makes it possible to obtain the ratio of these parameters and compare data between cells from different sources. The authors tried to uncover the mechanism of SOCE currents in cardiomyocytes. Specific inhibitors were used to identify the channels involved. Western blot analysis showed different levels of corresponding proteins expression in ventricular and atrial cardiomyocytes, which explained the differences in SOCE currents in these cell types.
As a small note:
Information about the lack of knowledge about SOCE currents in cardiomyocytes should be added to the introduction in order to further interest the reader.
Reviewer 2 Report
Comments and Suggestions for Authors
I had the pleasure reviewing the present study by Hermes et al. aimed to investigate the underlying mechanisms of store-operated calcium entry (SOCE) in the context of nuclear Calcium and potentially calcium/calmodulin-dependent signaling cascades. The study showed that both TRPC and Orai channels may contribute to SOCE in adult cardiomyocytes, and that SOCE in atrial and ventricular myocytes is able to cause robust calcium increases in the nucleus. It also contrasts atrial and ventricular relative contribution of SOCE. The work is novel and employs adequate methods to answer the respective research questions. Even upon careful review, no major concerns/flaws were detected and I congratulate the authors to a comprehensive manuscript. However, the authors draw the conclusion that SOCE alters CAMK/NFAT signaling, is there any direct experimental evidence e.g. in detubulated cells for this apart from the fact that SOCE seems to affect nuclear Calcium? The authors also might want to comment on the kinetics of altered nuclear Ca due to SOCE in the context of normal excitation contraction coupling.
